# Novel biomarker profiles to improve individual diagnosis and prognosis in patients with suspected inflammatory bowel disease: protocol for the Nordic inception cohort study (NORDTREAT)

Anja Fejrskov [ID],[1,2,3] Johannes David Füchtbauer [ID],[1,4,5] Lóa G Davíðsdóttir,[6] Jonas Halfvarson,[7] Marte Lie Høivik,[8,9] Michael Dam Jensen,[10,11] Joachim Høg Mortensen,[12] Lene Nyholm Nielsen,[13] Martin Rejler [ID],[14,15] Dirk Repsilber,[16] Johan D Söderholm,[17] Claus Aalykke,[5] Vibeke Andersen,[2,11,18,19] Robin Christensen,[3,20] Jens Kjeldsen [ID] [1,4]

AF and JDF are joint first authors.

**Correspondence to**
Anja Fejrskov; anja.fejrskov@rsyd.dk and Dr Johannes David Füchtbauer; David.Fuchtbauer@rsyd.dk

## ABSTRACT

**Introduction** Inflammatory bowel disease (IBD), including ulcerative colitis and Crohn's disease, can be challenging to diagnose, and treatment outcomes are difficult to predict. In the NORDTREAT cohort study, a Nordic prospective multicentre study, we aim to identify novel molecular biomarkers of diagnostic value by assessing the diagnostic test accuracy (cross-sectionally), as well as the prognostic utility when used as prognostic markers in the long-term (cohort study). In the diagnostic test accuracy study, the primary outcome is a successful diagnosis using one or more novel index tests at baseline compared with the ECCO criteria as the reference standard. The composite outcome of the prognostic utility study is 'severe IBD' within 52 weeks from inclusion, defined as one or more of the following three events: IBD-related surgery, IBD-related hospitalisation or IBD-related death.

**Methods and analysis** We aim to recruit 800 patients referred on suspicion of IBD to this longitudinal observational study, a collaboration between 11 inclusion sites in Denmark, Iceland, Norway and Sweden. Inclusion will occur from February 2022 until December 2023 with screening and baseline visits for all participants and three outcome visits at weeks 12, 26 and 52 after baseline for IBD-diagnosed patients. Biological material (blood, faeces, biopsies, urine and hair), clinical data and lifestyle information will be collected during these scheduled visits.

**Ethics and dissemination** This study will explore novel biomarkers to improve diagnostic accuracy and prediction of disease progression, thereby improving medical therapy and the quality of life for patients with IBD.
The study is approved by the Ethics Committee (DK: S-20200051, v1.4, 16.10.2021; IS: VSNb2021070006/03.01, NO: 193064; SE: DNR 2021-05090) and the Danish Data Protecting Agency (20/54594). Results will be disseminated through peer-reviewed journals, patient associations and presentations at international conferences.

**Clinical trial registration number** NCT05414578; Pre-results.

## STRENGTHS AND LIMITATIONS OF THIS STUDY

⇒ All participants included on suspicion of inflammatory bowel disease and treatment naïve.
⇒ Strong international collaboration and extensive inclusion area with 11 sites in four Nordic countries.
⇒ Prospective, longitudinal study design with rigorously collected biological material and data on disease activity and lifestyle.
⇒ Patient heterogeneity (four countries).
⇒ Ascertainment bias; as some patients with severe disease course may not be eligible for inclusion if acute treatment is initiated.

## INTRODUCTION

Inflammatory bowel disease (IBD), primarily ulcerative colitis (UC) and Crohn's disease (CD) are idiopathic disorders characterised by chronic gastrointestinal inflammation and relapsing symptoms like diarrhoea, abdominal pain and fatigue. A large proportion of the patients experience severe disease manifestations like bowel obstruction, fistulising disease, liver disease and colorectal cancer. Additionally, extraintestinal manifestations occur in 15%–40%.[1] Consequently, hospitalisations, surgical interventions, recurring sick leave and life-long medication severely impact the lives of patients and families. Even though the incidence of IBD appears to have reached a plateau in most Western European countries, it is rapidly increasing in many newly industrialised countries.[2] The prevalence of IBD has increased substantially in the last three decades, exceeding 0.5% in many European countries.[2–4] An estimated 1.3 million people in Europe suffer from IBD, imposing

a substantial health, social and economic burden.[1] There is a need to improve the management of IBD.

Diagnosis of IBD can be cumbersome.[5–7] Furthermore, IBD is a heterogeneous disease and its course differs substantially between patients. The lack of vigorous biomarkers for individual disease characterisation potentially leads to delayed diagnosis, organ damage, worsened outcome, increased mortality and an ever-increasing disease burden.[8] Personalised medicine (PM) could be a suitable strategy for optimising both diagnostics and treatment of IBD.[9] The vision of the International Consortium for Personalised Medicine (ICPerMed) describes 'an approach to medicine that integrates an individual's characteristics for early disease diagnosis, prognosis, optimal choice of treatment, accurate disease risk estimation, and targeted prevention'.[10] In gastrointestinal diseases, PM is based on advancements in biomarker detection in a variety of omics-platforms, including genomics, transcriptomics, microbiomics, proteomics and metabolomics, as well as body imaging methods, for example, CT scans and MRI scans. Recent advances led to the identification of several potential biomarkers for diagnosis and treatment response in IBD.[11–17] However, they are not in use in clinical practice. Even though early diagnosis and efficient treatment strategies for IBD are cornerstones for reducing irreversible organ damage, identifying biomarkers that can efficiently support clinical decision-making is still warranted.

This protocol article introduces a prospective clinical study that aims to identify novel diagnostic biomarkers for more accurate early diagnosis of IBD and assess the utility of novel prognostic biomarkers in clinical outcomes with various disease manifestations. Notably, the study evaluates biomarkers in a relevant clinical setting, that is, among patients referred to the hospital on suspicion of IBD and diagnosed according to the ECCO criteria.[5]

### Aims

This study aims, on the one hand, to improve accuracy in diagnosing IBD in the form of a *diagnostic test accuracy study*. The main hypothesis is that biochemical markers can differentiate individuals referred to a gastroenterology department with suspected IBD into individuals with (IBD) and without (non-IBD) diagnosis. The primary aim is to identify molecular biomarkers (eg, from blood and stools) for discriminating between individuals diagnosed with IBD (CD, UC and unclassified inflammatory bowel disease (IBD-U)) and non-IBD. Secondary aims include investigating whether clinical, genetic, microbiome and lifestyle information and combinations thereof (eg, gene-environment interaction analyses) are associated with IBD diagnosis.

On the other hand, the main hypothesis of the *prognostic utility study* is that prognostic factors can distinguish different pathways of disease progression. The primary aim of this study is to identify prognostic biomarkers that can define different IBD disease courses. The secondary aims are to evaluate whether including information on clinical and lifestyle factors and combinations thereof can improve prognostic value.

## METHODS AND ANALYSES

### Design

The NORDTREAT cohort study is a Nordic multicentre collaboration on diagnostic factors and prognostic factors of IBD among patients referred to the hospital with suspected IBD. All referred patients will be carefully evaluated to ensure that patients with suspected IBD are invited to participate in the study at their first clinic visit. Informed consent will be obtained at the screening visit (week −4). After study material collection (ie, blood, stool, biopsies, hair, urine, questionnaires and patient-reported outcome measures (PROMs)) and diagnostic evaluation, IBD diagnosis will be established or rejected at the baseline visit (week 0). IBD patients will be followed until week 52 with three additional visits (weeks 12, 26 and 52), whereas non-IBD patients will complete questionnaires at baseline visit and week 52 (figure 1).

The prospective study consists of a cross-sectional part with focus on diagnosis and a longitudinal cohort study (52 weeks) with focus on prognosis (ie, a prognostic utility study). The inclusion of patients into the NORDTREAT cohort study will be reported as a flowchart (figure 2).

The objective of the *diagnostic test accuracy study* is to determine the sensitivity and specificity of biomarkers of interest (baseline measurements, see Biomarkers section) to diagnose IBD when applying the ECCO diagnostic criteria as the reference standard.[5]

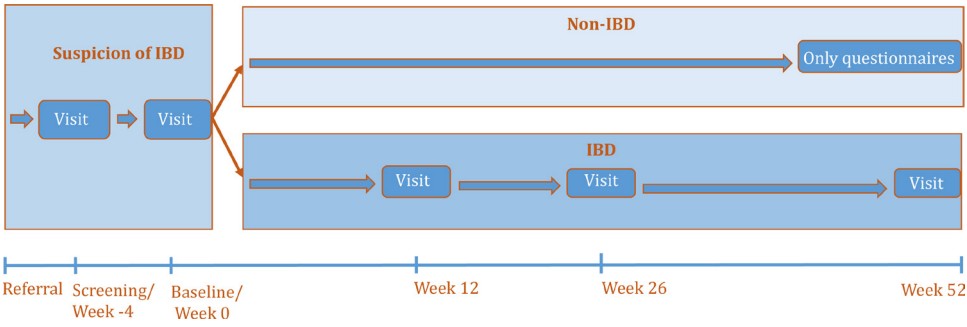

**Figure 1** Schedule for patients with a suspected IBD diagnosis.

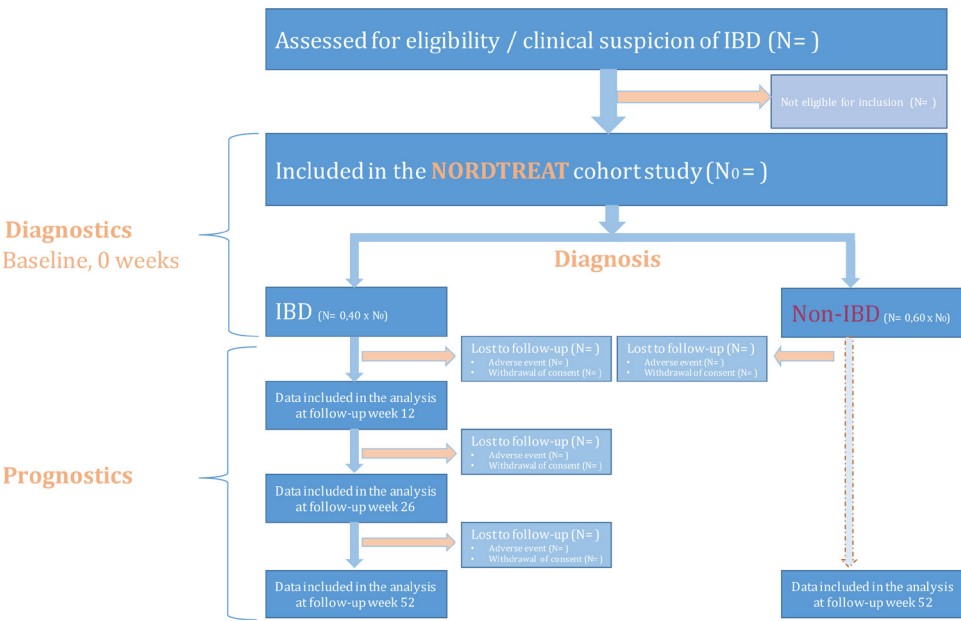

**Figure 2** Flowchart for patients assessed eligible for inclusion in the NORDTREAT cohort study.

The objective of the *prognostic utility study* is to examine whether the biomarkers of interest predict severe disease outcomes in patients with IBD. Severe IBD is defined as those experiencing IBD-related surgery, IBD-related hospitalisation (except planned procedures) or IBD-related death within 52 weeks of inclusion.

### Setting

This Nordic multicentre study is a collaboration between gastroenterology departments in 11 Nordic hospitals (online supplemental appendix 1). The study enrolment will occur between 7 February 2022 and 31 December 2023. The first patient's first visit was conducted in Oslo University Hospital in February 2022, and the first patient's last visit was in February 2023. All inclusion sites will conduct their last patient's first visit and last patient's last visit in December 2023 and December 2024, respectively.

### Patient characteristics and eligibility criteria

Inclusion criteria: patients≥18 years of age referred to a gastroenterology department with suspected IBD who give written informed consent to participate in the study. Exclusion criteria: patients with a previous diagnosis of CD, UC or IBD-U or patients unable to provide informed consent or comply with protocol requirements, for example, due to recreational drug or alcohol misuse.

The clinical data and lifestyle information collected for the study will be presented in tabular form (see table 1) and include the following.

### Clinical interview

Demographics (sex, age, marital status, educational level, occupational status and smoking), disease history (previous disease including debut of gastrointestinal symptoms) and surgery (abdominal and perianal), occurrence of intestinal complications (defined as strictures

and fistulas), occurrence of perianal disease, extraintestinal manifestations, medication (past and present), family history (of IBD and colorectal cancer) and at the week 52 visit: relapses and surgery (abdominal and perianal) are recorded.

### Clinical examination

Includes a perianal examination of study subjects with perianal CD (accounting for the number of draining and non-draining perianal fistula).

### Vital signs

Include measurement of height, weight, heart rate, blood pressure, and temperature.

### Endoscopy

Standardised scoring systems will record macroscopic findings from the ileocolonoscopy: for patients with UC, the Endoscopic Mayo Score per segment is applied, that is, ascending, transverse, descending, sigmoid colon and rectum.[18] For patients with CD, the Simple Endoscopic Score for Crohn's Disease (SES-CD) per segment is applied, that is, terminal ileum, ascending, transverse, left colon and rectum.[19] Patients diagnosed with IBD-U at baseline will be assessed with the Endoscopic Mayo Score per segment and the SES-CD per segment. The endoscopy is recorded on video (approximately 20s per segment). Furthermore, standard routine biopsies from the ileum, ascending, transverse, descending colon, and rectum will be obtained.

### Medical records

Routine clinical information, clinical findings and procedures (endoscopic and surgical), test results (ie, histology, radiology (including small bowel MRI, capsule endoscopy, CT scan and ultrasonography), microbiology and biochemistry), diagnosis, treatment and complications

**Table 1** Overview of clinical and lifestyle information collected according to visits (weeks after diagnosis)

| | IBD | | | | non-IBD | |
|---|---|---|---|---|---|---|
| Week | −4/0 | 12 | 26 | 52 | −4/0 | 52 |
| Clinical information | | | | | | |
| Clinical interview and examination | x | x | | x | x | |
| Vital signs | x | x | x | x | x | |
| Endoscopy | x | | | x | x | |
| Medical records | x | x | | x | x | |
| Montreal classification | x | | | x | | |
| Lifestyle information, disease activity index and PROMs | | | | | | |
| SF36 | x | x | x | x | | |
| IBDQ | x | x | x | x | | |
| EQ-5D-5L | x | x | x | x | | |
| IBD-F | x | x | x | x | x | x |
| HBI (CD only) | x | x | x | x | | |
| Partial Mayo score (UC only) | x | x | x | x | | |
| ROME IV | x | x | x | x | x | x |
| Diet | x | | | x | x | x |

CD, Crohn's disease ; EQ-5D-5L, EuroQol 5 Dimension 5 Level; HBI, Harvey-Bradshaw Index; IBD, inflammatory bowel disease; IBD-F, Inflammatory Bowel Disease Fatigue (part 1); IBDQ, Inflammatory Bowel Disease Questionnaire; PROMs, patient-reported outcome measures; SF36, Short Form 36; UC, ulcerative colitis .

including hospitalisations will be retrieved from the medical records.

### Montreal classification
Based on results from the routine examinations, the study subjects will be classified according to the internationally accepted IBD classification system, the Montreal Classification.[20]

### Lifestyle information, Disease Activity Index, disease questionnaires and PROMs
Participants will complete validated questionnaires (on paper or electronically) on disease activity, quality of life, including *EQ-5D-5L*[21] (measure of, eg, mobility, personal care, usual activities, pain/discomfort and anxiety/depression), *SF-36*[22] (including perception of health, limitations in daily and social activities, general mental health (psychological distress and well-being), and vitality (energy and fatigue) within the last 4 weeks), *IBDQ*[23 24] (disease impact on patients daily functioning and quality of life, eg, abdominal pain, sleep, incontinence, ability to work and conduct social activities within the last 2 weeks), *IBD-F*[25] (disease-specific questionnaire comprising five questions about the severity of fatigue (part 1) and *ROME IV*[26] (The Rome IV Diagnostic Questionnaire for Functional Gastrointestinal Disorders in Adults). The electronic case report form (eCRF) platform Viedoc (Viedoc Technologies, Uppsala Sweden) is applied, and in Denmark REDCap (Vanderbilt University, Nashville, TN, USA) is used for diet questionnaires.

Before each visit, patients with IBD will register clinical disease activity, Harvey-Bradshaw Index (*HBI*)[27 28] (diary for patients with CD including two clinical parameters: abdominal pain and number of liquid or soft stools), or *Partial Mayo Score* (diary for patients with UC or IBD-U including the two non-invasive components of the full Mayo Score (stool frequency and rectal bleeding)). All participants will perform diet questionnaires at baseline and week 52.[29]

### Biological material
Blood, biopsies, faeces, hair and urine will be collected at the scheduled visits (table 2).

At baseline, all study subjects have blood samples (whole blood (4 mL), plasma (10 mL), serum (20 mL) and Paxgene blood RNA system (2,5 mL)) and biopsies collected (eight biopsies from predefined locations collected in Allprotect reagent and six biopsies from predefined locations snap frozen in liquid nitrogen or FlashFREEZE (Milestone Medical, product no. 100 780 from Axlab A/S, DK), ensuring a standardised freezing procedure at −80°C). Furthermore, the faecal samples collected at the patient's home will consist of: one stool collection tube with RNALater, DNA stabiliser and a dry sample, respectively. Additionally, a tuft of hair and urine (8 mL) will be collected.

Blood samples will not be collected on the day of the ileocolonoscopy, and faecal samples will not be collected during the colonic cleansing to minimise any potential influence from the standard cleansing procedure conducted before an endoscopy.

**Table 2** Overview of the biological material collected in the study

| Biological material | Specifications | Baseline<br>IBD and non-IBD | Week 12<br>IBD | Week 26<br>IBD | Week 52<br>IBD |
|---|---|---|---|---|---|
| Blood | Whole blood (EDTA-K2) | x | | | |
| | Plasma (EDTA-K2) | x | x | x | x |
| | Serum | x | x | x | x |
| | Paxgene | x | | | x |
| Biopsies | Allprotect reagent | x | | | x |
| | Snap frozen | x | | | x |
| Faecal samples | DNA stabiliser | x | x | x | x |
| | Dry | x | x | x | x |
| | RNALater | x | x | x | x |
| Hair | 10–20 mg | x | | | x |
| Urine | Spot urine | x | | | x |

IBD, inflammatory bowel disease.

On follow-up visits at week 12 and week 26, all patients diagnosed with IBD will have blood samples (plasma (10 mL) and serum (20 mL)) and faecal samples (one stool collection tube with RNALater, DNA stabiliser and a dry sample, respectively) collected. At the week 52 visit for IBD patients, biological material similar to baseline will be collected (except for whole blood). The biological material will be stored at local facilities according to national procedures, and accordingly available for future research projects on approval of the NORDTREAT executive committee. Standard operating procedures with detailed information can be found in online supplemental appendix 2.

### Primary endpoint(s)

The primary endpoint for the cross-sectional *diagnostic test accuracy study* is diagnosis according to the reference standard from ECCO criteria,[5] that is, IBD (including CD, UC, IBD-U) or non-IBD (patients not diagnosed with IBD) at referral.

The primary endpoint for the longitudinal *prognostic utility study* is a composite outcome measure. We will include multiple (three mutually associated) endpoint components combined as a Boolean function; this will be our primary outcome measure derived from randomised trials while also anticipating that the approach will be associated with increased statistical efficiency (eg, power). The composite outcome consists of the proportion of patients who experience severe IBD over 52 weeks after inclusion, defined as either IBD-related surgery, IBD-related hospitalisation (except planned procedures) or IBD-related death.

### Key secondary endpoints in the prognostic utility study

Major secondary outcomes assessed and analysed 52 weeks from baseline will include IBD-related surgery (number (no.) in %), IBD-related hospitalisation (no., %), IBD-related deaths (no., %) as well as non-corticosteroid-free

(local acting systemic administered (budesonide) and systemic steroids (prednisolone)) clinical remission at week 12.[30] Clinical remission in CD is defined as patient-reported outcome 2 (abdominal pain≤1 and stool frequency≤3) or Harvey Bradshaw Index<5. Clinical remission in UC is defined as patient-reported outcome 2 (rectal bleeding=0 and stool frequency=0) or Partial Mayo score (<3 and no score>1). Non-corticosteroid-free endoscopic remission is assessed at week 52,[30] where endoscopic remission in CD is defined as Simple endoscopic score<3 points or absence of ulcerations (eg, SES-CD ulceration subscore=0). Endoscopic remission in UC is defined as Mayo endoscopic score=0 points or UC endoscopic index of severity≤1 point. Disease impact on patient's life will be assessed by the 32-item version of Inflammatory Bowel Disease Questionnaire (IBDQ) score (number (%) ≤ 170) and the numerical change of the SF36 subscales.

### Other secondary endpoints in the prognostic utility study

The composite secondary outcome is the proportion of patients who experience IBD complications within 52 weeks after inclusion, defined as the cumulative number of patients who experience one of the following (1) need for medication (immunomodulators, biologics or steroids) between diagnosis and 1 year, (2) need for surgery between diagnosis and 1 year, (3) need for hospitalisation between diagnosis and 1 year or (4) relapse between diagnosis and 1 year. Furthermore, the cumulative number of flares, adverse drug reactions, numerical change in the Fatigue Questionnaire, and cumulative glucocorticoid exposure will be assessed. Faecal calprotectin>250 µg/g (no., %), time to start of biological therapy, time to IBD-related surgery or hospitalisation and CD-related surgery (stenosis) will also be measured.

## Mid-term and long-term complications in the prognostic utility study

IBD-related hospitalisations will be assessed and defined as a composite endpoint combining the total number of IBD-related hospitalisations (including readmissions) and the cumulative length of hospital stay. Bowel damage in patients with CD (Lémann index for patients with an MRI), UC-related surgery (colectomy), CD-related surgery (no., %), IBD-related hospitalisations (no., %) and change in disease extent in UC (with a focus on macroscopic proximal disease extent) will be assessed among the mid-term complications. Among the long-term complications, the following three endpoints will be measured: gastrointestinal and extraintestinal dysplasia (no., %), cancer (gastrointestinal and others in no., %) and mortality (no., %).

## Biomarkers

At the time of submission, the NORDTREAT executive committee has approved projects on the following topics in relation to diagnosis and prognosis (including inflammatory intensity, disease extend, tissue damage, tissue repair activity and severe disease course): extracellular matrix proteins, microbiome, mycobiome, faecal markers, genetic risk factors, metabolomics antibodies towards microbial and autoantigens.

## Statistical methods

Referring to earlier studies, approximately 40% of a population referred with suspected IBD will have the diagnosis confirmed according to the ECCO criteria (or comparable).[5 31] Cost and practical issues will restrict the sample size to approximately 800 patients, with an expected 80% participation rate.

In the *diagnostic test accuracy study*, we will compare two independent binomial proportions, using the likelihood ratio (LR) statistic with a $\chi^2$ approximation and two-sided significance level (0.05). With an expected sample size of 640 participants (assuming a biomarker positivity-to-negativity ratio of 1 to 1), we will reach an approximate statistical power of 100%, assuming that the proportion of patients diagnosed with IBD is 80% in the biomarker positive group and 20% in the biomarker negative group. If we include 640 study participants, we will potentially ('rule of thumb') be able to build a multivariable statistical model with as many as 64 covariates. Thus, we will examine a variety of biochemical variables for diagnosing IBD (see Biomarkers section). The expectation is that a proportion of 100–200 preliminary biomarkers will be evaluated in this model to distinguish non-diagnostic markers and preliminary biomarkers that can be replicated. When applying a significance level of 0.05, each biomarker has a 5% risk of being replicated by chance. Thus, in the evaluation of, for example, 100–200 biomarkers, five to 10 biomarkers may be replicated by chance. In contrast, the risk of not replicating 'true' biomarkers with low effects is estimated to be approximately 20%. All replicated biomarkers will be built into a prediction model that will be optimised and evaluated by cross-validation. The primary hypothesis is that the biomarker signature will be able to characterise the participant population into an IBD and a non-IBD group.

In the *prognostic utility study*, we expect a sample size of 256 IBD patients and a 10% event rate for severe IBD within 52 weeks.[32 33] If we assume a 15% event rate for the biomarker positive group and a 5% event rate for the biomarker negative group, and anticipate an even 1:1 biomarker positive-to-negative distribution, approximately 80 patients in each group will increase the statistical power above the 90% threshold.

## Comparison of measures of diagnostic accuracy

The basic measures of quantifying the diagnostic accuracy of a test will be the diagnostic OR, sensitivity and specificity measures. From these measures, we will calculate the LR, the ratio of the probability of the index test result among patients who have IBD to the probability of the same test among patients who do not have IBD (non-IBD). Since all diagnostic tests will be performed on each patient, paired data results and methods that account for the correlated binary outcomes are necessary (eg, McNemar's test).

## Prognostic factor research (the prognostic utility study)

The primary statistical analysis will be based on the proportion of participants experiencing severe IBD among biomarker-positive versus biomarker-negative patients. The efficacy of the prognostic biomarker(s) will be considered as the proportion of the patients with different disease outcomes among the biomarker-positive patients compared with the biomarker-negative patients. The associations of the suspected important molecular biomarkers with other variables will be tested with nonparametric tests: with Spearman's rank correlation (rs) for continuous variables and the Wilcoxon rank-sum test or Kruskal-Wallis test, including a Wilcoxon-type test for trend across ordered groups where appropriate, for categorical variables.

Logistic regression models will be utilised with individual markers as the exposure variables and composite clinical severity as the outcome (dependent variable). The analyses will be adjusted simultaneously for sex, age and prescribed therapy. Potential interactions between biomarker status (positive/negative) and specific drug type will be evaluated too. Covariates comprise of various measures within genetics, transcriptomics, microbiomes and proteomics (see Biomarkers section). We will attempt to develop prediction models containing several conventional risk factors (ie, age, sex and type of drug therapy) with or without any novel biomarkers and calculate improvements in predictive ability using risk discrimination and reclassification measures. We will use a two-stage approach that allows for the examination of centre heterogeneity through the calculation of the C-index, a measure of risk discrimination and changes within each centre before pooling results.

We will attempt to develop multivariable prediction models, relating multiple predictors for a particular individual to the probability of or risk for the presence (diagnosis of IBD at baseline) or future occurrence (prognosis) of a particular outcome, such as intestinal surgery within the first year. Predictors such as biomarkers are covariates, explored as prognostic or predictive factors. Models will be built using regularised regression methods, such as sparse Partial Least Squares (PLS) regression or Least Absolute Shrinkage and Selection Operator (LASSO), as well as tree-based machine learning approaches, such as random forest, or non-linear non-symbolic deep learning approaches, capitalising on available published data sets. Also, multiple data sets will be used for integrative analyses, to explore if combinations or interactions of variables of different kinds could aid predictive or prognostic potential of our models.

## Patient and public involvement

The NORDTREAT project has evolved in collaboration with patient research partners (PRPs) from the Danish Colitis-Crohn's Association, the Norwegian Mage-tarmforbundet, the Swedish Mag- och tarmförbundet and from Icelandic patient representatives ensuring implementation of patient perspectives in aspects such as, for example, research aims and outcomes. The PRPs were involved before the formal project start (adding value in the design of the study) and they are continuously informed on study progress. Overall, the PRPs add value in the project conduct through their engagement in the Patient Advisory Council (online supplemental appendix 3) increasing knowledge-sharing, awareness on the project and research comprehensiveness.

When reporting the *diagnostic test accuracy study*, we will adhere to the STARD guideline[34] and the TRIPOD statement,[35] and when reporting the *prognostic utility study*, we will apply the STROBE[36] as well as the TRIPOD statement.[35] Before any analyses are initiated, an elaborated statistical analysis plan indicating explicit figures and tables will be provided.

## Project organisation

The NORDTREAT partnership builds on extensive professional experience, a strong track record of collaborative clinical projects and participation in a number of European and global consortia (eg, the IBD Character Consortium, the SYSCID Consortium and the International IBD Genetic Consortium).

The NORDTREAT cohort study is conducted in parallel and as part of the NORDTREAT trial (a biomarker-stratified randomised controlled trial (RCT)). The aim of the NORDTREAT RCT is to demonstrate that personalised therapy can be applied to patients with IBD, by assessing if a top-down treatment can improve clinical outcomes in IBD patients with an increased risk of poor disease course (https://clinicaltrials.gov/ct2/show/NCT05180175).

The operational organisation of the NORDTREAT study is highlighted in online supplemental appendix 3. The *Steering Committee* (SC), whose members are responsible for planning and organising the study and scientific follow-up, includes the leaders of the six predefined work packages (WPs): WP1: Clinical phenotyping and biobanking (Professor Marte Lie Høivik), WP2: NORDTREAT trial (Professor Jonas Halfvarson), WP3: Host and gut microbiome signatures (NORDTREAT cohort) (Professor Vibeke Andersen), WP4: Bioinformatics and integrated analyses (Professor Dirk Repsilber), WP5: Exploitation and dissemination (Professor Johan D Söderholm) and WP6: Project management (Professor Jonas Halfvarson). Elected SC members also include Professor Jens Kjeldsen, Professor Lóa G Davíðsdóttir, Professor Einar S Björnsson, Professor Asle W Medhus, associate Professor Johannes R Hov, consultant PhD Vendel Kristensen, associate Professor Randi Opheim, PhD Joachim Høg Mortensen, PhD fellow David Füchtbauer, and PhD fellow Anja Fejrskov. The Steering Committee conducts regular meetings and has feedback from the *External Advisory Board* composed of internationally renowned representatives of academia and the biomedical industry: Professor Jack Satsangi, Oxford University (GB), Professor Philip Rosenstiel, Kiel University (DE), Professor Mauro D'Amato, Karolinska Institute (SE), vice president at Q-linea Mats Gullberg (SE) and from PRPs in the national patient organisations (*Patient Advisory Council* (online supplemental appendix 3)).

Finally, the *Executive Committee* (members listed in online supplemental appendix 3) is responsible for project coordination and management during the research project's preparation, operation and finalisation. Biweekly online meetings are conducted, ad-hoc meetings when necessary and face-to-face meetings are held twice a year.

## Perspectives

Substantial differences exist between patients suspected of IBD when comparing symptoms, onset, prognosis and treatment, which makes a one-size-fits-all approach insufficient. With current knowledge and clinical methodologies, the diagnosis and prognosis of IBD is challenging without invasive investigations. However, newly identified biomarkers may have implications for future disease management, as they could be used to differentiate patients with IBD and patients with other comparable gastrointestinal symptoms or predict disease outcomes. If we succeed in identifying diagnostic or prognostic biomarkers of IBD, substantial improvement will be given to IBD treatment, including reduced side-effects and complications, increased quality of life for patients and their relatives and a reduced socioeconomic burden on society. Relevant findings with potential clinical value will be evaluated and replicated in other prospective cohorts if possible.

## Ethics and dissemination

The study is approved by the Ethics Committee (DK: S-20200051, v1.4, 16.10.2021; IS: VSNb2021070006/03.01, NO: 193064; SE: DNR 2021-05090) and the Danish Data Protecting Agency (20/54594). The procedures followed are in accordance with the ethical standards of the responsible committee on human experimentation (institutional and national) and with the Helsinki Declaration of 1975 with later amendments.

The target journal for the primary outcome will be a gastroenterology medical journal. Subsequently, other hypotheses will be tested, and manuscripts will be prepared with the intention of submitting to specialised journals in microbiome research, immunology, genetics, nutrition and gastroenterology. In addition to the scientific reporting, key findings with translational implications will be communicated to health professionals, patient organisations, public health policymakers and the public through various media and news activities. When disseminating the results, we will apply the recommendations of the *International Committee of Medical Journal Editors (ICMJE).*

**Author affiliations**
[1]Department of Medical Gastroenterology, Odense University Hospital, Odense, Denmark
[2]Department of Internal Medicine, Molecular Diagnostics and Clinical Research Unit, Institute of Regional Health Research, University Hospital of Southern Denmark, Aabenraa, Denmark
[3]Section for Biostatistics and Evidence-Based Research, Parker Institute, Frederiksberg, Denmark
[4]Research Unit of Medical Gastroenterology, Department of Clinical Research, University of Southern Denmark, Odense, Denmark
[5]Internal Medicine and Emergency Department, Odense University Hospital, Svendborg, Denmark
[6]Department of Gastroenterology, Landspitali National University Hospital of Iceland, Reykjavik, Iceland
[7]Department of Internal Medicine, Örebro University Hospital, Örebro, Region Örebro län, Sweden
[8]Department of Gastroenterology, Oslo University Hospital, Oslo, Norway
[9]University of Oslo Institute for Clinical Medicine, Oslo, Norway
[10]Department of Internal Medicine–Gastroenterology, Lillebaelt Hospital, University Hospital of Southern Denmark, Vejle, Denmark
[11]Institute of Regional Health Research, Faculty of Health Sciences, University of Southern Denmark, Odense, Denmark
[12]Gastrointestinal Diseases, Nordic Bioscience A/S, Herlev, Denmark
[13]Research Unit of Medical Gastroenterology and Hepatology, Hospital South West Jutland, Esbjerg, Denmark
[14]Jönköping Academy for Improvement in Health and Welfare, Jönköping University, Jönköping, Sweden
[15]Futurum-Academy for Healthcare, Futurum Academy of Health and Care, Jönköping, Region Jönköping County, Sweden
[16]School of Health and Medical Sciences, Örebro University, Örebro, Sweden
[17]Department of Biomedical and Clinical Sciences, Linköping University, Linköping, Östergötland, Sweden
[18]Institute of Molecular Medicine, University of Southern Denmark, Odense, Denmark
[19]OPEN, Open Patient data Explorative Network, University of Southern Denmark, Odense, Denmark
[20]Research Unit of Rheumatology, Department of Clinical Research, University of Southern Denmark, Odense, Denmark

**Acknowledgements** Acknowledgements are based on the *ICMJE* recommended criteria for authorship. Professor Björnsson E S, Professor Carlson M, Professor Heitmann B L, associate Professor Hov J R, CEO in Bio-Me Isaksen M, Professor Kruse T A, Professor Moum B, associate Professor Ophei R, MD, PhD Ricanek P and senior coordinator and innovation advisor Wallin Å contributed to the design of the study. The Open Patient data Explorative Network (OPEN), Odense University Hospital, Region of Southern Denmark has facilitated storage and documentation of the biological material collected in the OPEN biobank framework. Thanks to research advisor, MPH Caroline Moos for English text editing. Thanks to the Danish Colitis-Crohn's Association, the Norwegian Mage-tarmforbundet, the Swedish Mag-och tarmförbundet and the Icelandic patient representatives for valuable advice when planning and conducting the NORDTREAT study.

**Contributors** AF (0000-0002-1257-8630), JDF (0000-0002-9591-2963), VA (0000-0002-0127-2863), RC (0000-0002-6600-0631) and JK (0000-0001-8148-6572) wrote the first draft. LGD, JH (0000-0003-0122-7234), MLH (0000-0002-0104-465X), JHM (0000-0003-0326-7264), MR (0000-0003-3286-2898), DR (0000-0002-7173-5579), JDS (0000-0002-3250-5367), VA, RC and JK contributed to the conception and design of the study and are all members of the Executive Committee. MDJ (0009-0004-8432-1764), LNN and CA are principal investigators and contributed to the inclusion of patients in the Region of Southern Denmark. All authors accepted the final submitted version.

**Funding** The NORDTREAT project has received funding from NordForsk (30 000 000 SKR) via Vinnova (grant number 2019-01185 NORDTREAT), Rannis (NordForsk no. 90569-grant no. 199782-0611), The Research Council of Norway (grant no. 2988039) and Innovation Fund Denmark (90569 NORDTREAT: grant number 8114-00026B). The Danish partners have allocated the funding (4 800 000 DKR) for remuneration of project staff, biobanking and analyses of biological material. Furthermore, the University Hospital of Southern Denmark, Hospital Sønderjylland has funded 1 125 000 DKR for PhD salary (Fejrskov A (grant number not applicable)), The Region of Southern Denmark (Fri og Strategisk Forskning) has funded 1 137 000 DKR (Fejrskov A (J.no.: 21/17578 and Efond: 1084) and Füchtbauer JD (grant number A1381)) for biobanking and analyses of biological material. Knud and Edith Eriksen Memorial Fund has funded 75 000 DKR for PhD salary (Fejrskov A (case no. 62786-2022)). Odense University Hospital has funded 584 000 DKR for PhD salary (Füchtbauer JD (grant number A5031)). Section for Biostatistics and Evidence-Based Research, the Parker Institute, Bispebjerg and Frederiksberg Hospital (Christensen R) is supported by a core grant from the Oak Foundation (OCAY-13-309). None of the researchers participating in the project have any financial connection to the funding organisation, nor do they have a financial interest in the study.

**Competing interests** None declared.

**Patient and public involvement** Patients and/or the public were involved in the design, or conduct, or reporting or dissemination plans of this research. Refer to the Methods section for further details.

**Patient consent for publication** Not applicable.

**Provenance and peer review** Not commissioned; externally peer reviewed.

**ORCID iDs**
Anja Fejrskov http://orcid.org/0000-0002-1257-8630
Johannes David Füchtbauer http://orcid.org/0000-0002-9591-2963
Martin Rejler http://orcid.org/0000-0003-3286-2898
Jens Kjeldsen http://orcid.org/0000-0001-8148-6572

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
