## [Reviewer comments · BMJ Open]

WPEER REVIEW HISTORY

BMJ Open publishes all reviews undertaken for accepted manuscripts. Reviewers are asked to complete a checklist review form (<http://bmjopen.bmj.com/site/about/resources/checklist.pdf>) and are provided with free text boxes to elaborate on their assessment. These free text comments are reproduced below.

ARTICLE DETAILS

TITLE (PROVISIONAL)	Novel Biomarker Profiles to Improve Individual Diagnosis and Prognosis in Patients with Suspected Inflammatory Bowel Disease: Protocol for the Nordic Inception Cohort Study (NORDTREAT)
AUTHORS	Fejrskov, Anja; Füchtbauer, Johannes David; Davíðsdóttir, Lóa; Halfvarson, Jonas; Høivik, Marte; Jensen, Michael; Mortensen, Joachim; Nielsen, Lene; Rejler, Martin; Repsilber, Dirk; Söderholm, Johan; Aalykke, Claus; Andersen, Vibeke; Christensen, Robin; Kjeldsen, Jens

VERSION 1 – REVIEW

REVIEWER	Pritzker, Kenneth University of Toronto
REVIEW RETURNED	02-Feb-2024

GENERAL COMMENTS	The authors will be compiling extensive data on IBD patients. Suggestions to optimize outcomes from this study: 1. Search for new Diagnostic criteria Beyond differential diagnosis, (IBD, yes or no), this might include other features such as inflammation intensity; repair activity, repair extent, recurrence of acute disease probability .2. Statistical Analysis: Suggest that artificial intelligence/machine learning techniques be explored and compared to conventional statistical analysis.3. If not already in the protocol , suggest that sera be conserved and be made available to future investigators studying IBD biomarkers.
--

REVIEWER	Sakurai, Toshiyuki The Jikei University School of Medicine
REVIEW RETURNED	06-Mar-2024

GENERAL COMMENTS	Review of article bmjopen-2023-083144 Novel Biomarker Profiles to Improve Individual Diagnosis and Prognosis in Patients with Suspected Inflammatory Bowel Disease: Protocol for the Prospective NORDTREAT Cohort Study by Anja Fejrskov, et al. General comments This is the protocol of the multicenter prospective cohort study trying to identify novel molecular biomarkers for the diagnosis and prognosis of IBD.
--

	This protocol is well written and is suitable for acceptance, except for one major problem. In the section on “Biological materials” and “endpoints”, I can’t figure out what kind of biomarkers the authors try to identify. Is it miRNA, gut microbiota (16S rRNA Gene Amplicon Sequence), or others? I believe this is a comprehensive analysis of bio-samples. I suppose it is better to describe it. Specific comments None.
--	--

VERSION 1 – AUTHOR RESPONSE

Reviewer: 1

Dr. Kenneth Pritzker, University of Toronto

Comments to the Author:

The authors will be compiling extensive data on IBD patients.

Suggestions to optimize outcomes from this study:

1. Search for new Diagnostic criteria

Beyond differential diagnosis, (IBD, yes or no), this might include other features such as inflammation intensity; repair activity, repair extent, recurrence of acute disease probability.

We agree, from the explorative point of view, that this suggestion would increase the clinical value of our finding and have hence tried to describe our intend more clearly in the manuscript.

In our cohort setup we aim to achieve these secondary outcomes through the prognostic utility study which not only takes diagnosis into account but also our key secondary endpoints including endoscopic activity scores (Mayo score, SES-CD), Montreal classification, PROs, hospitalisation, surgery etc.

We, in fact, do plan to look at extracellular matrix proteins (ECM) as biomarkers of repair and damage, as well as microbiome related markers and metabolomics.

We have added the following information to the manuscript:

“At the time of submission, the NORDTREAT executive committee has approved projects on the following topics in relation to diagnosis and prognosis (including inflammatory intensity, disease extend, tissue damage, tissue repair activity, severe disease course): extra cellular matrix (ECM) proteins, microbiome, mycobiome, faecal markers, genetic risk factors, metabolomics anti-bodies towards microbial and auto-antigens.”

2. Statistical Analysis: Suggest that artificial intelligence/machine learning techniques be explored and compared to conventional statistical analysis.

We agree that Machine Learning/AI is essential in predictive modelling, and have clarified the text to reflect this:

“We will attempt to develop multivariable prediction models, relating multiple predictors for a particular individual to the probability of or risk for the presence (diagnosis of IBD at baseline) or future occurrence (prognosis) of a particular outcome, such as intestinal surgery within the first year. Predictors such as biomarkers are covariates, explored as prognostic or predictive factors. Models will be built using regularised regression methods, such as sparse PLS or LASSO, as well as tree-based

machine learning approaches, such as random forest, or non-linear non-symbolic deep learning approaches, capitalising on available published datasets. Also, multiple datasets will be used for integrative analyses, to explore if combinations or interactions of variables of different kinds could aid predictive or prognostic potential of our models.”

3. If not already in the protocol, suggest that sera be conserved and be made available to future investigators studying IBD biomarkers.

We have clarified our intend to store biologic material for future projects:

“The biological material will be stored at local facilities according to national procedures, and accordingly available for future research projects upon approval of the NORDTREAT executive committee.”

Reviewer: 2

Dr. Toshiyuki Sakurai, The Jikei University School of Medicine

Comments to the Author:

Review of article bmjopen-2023-083144

Novel Biomarker Profiles to Improve Individual Diagnosis and Prognosis in Patients with Suspected Inflammatory Bowel Disease: Protocol for the Prospective NORDTREAT Cohort Study by Anja Fejrskov, et al.

General comments

This is the protocol of the multicenter prospective cohort study trying to identify novel molecular biomarkers for the diagnosis and prognosis of IBD.

This protocol is well written and is suitable for acceptance, except for one major problem.

In the section on “Biological materials” and “endpoints”, I can’t figure out what kind of biomarkers the authors try to identify. Is it miRNA, gut microbiota (16S rRNA Gene Amplicon Sequence), or others? I believe this is a comprehensive analysis of bio-samples. I suppose it is better to describe it.

We acknowledge the point of Dr. Toshiyuki Sakurai, and included a “Biomarkers” section to define the planned biomarker projects on the cohort:

“At the time of submission, the NORDTREAT executive committee has approved projects on the following topics in relation to diagnosis and prognosis (including inflammatory intensity, disease extend, tissue damage, tissue repair activity, severe disease course): extra cellular matrix (ECM) proteins, microbiome, mycobiome, faecal markers, genetic risk factors, metabolomics anti-bodies towards microbial and auto-antigens.”

Specific comments

None.

VERSION 2 – REVIEW

REVIEWER	Pritzker, Kenneth University of Toronto
REVIEW RETURNED	14-Apr-2024
GENERAL COMMENTS	The authors have responded adequately to the reviewers' concerns.

REVIEWER	Sakurai, Toshiyuki The Jikei University School of Medicine
REVIEW RETURNED	24-Apr-2024

GENERAL COMMENTS	Thank you for your revised manuscript. All is addressed and this manuscript is suitable for publication.
--